# CITRIS: Causal Identifiability from Temporal Intervened Sequences

**Phillip Lippe,**[1] **Sara Magliacane,**[23] **Sindy Löwe,**[4] **Yuki M. Asano,**[1] **Taco Cohen,**[5] **Efstratios Gavves**[1]
[1]QUVA Lab, University of Amsterdam  [2]MIT-IBM Watson AI Lab
[3]INDE Lab, University of Amsterdam  [4]UvA-Bosch Delta Lab, University of Amsterdam
[5]Qualcomm AI Research[*]

## Abstract

We propose CITRIS, a variational framework that learns causal representations from temporal sequences of images with interventions. In contrast to the recent literature, CITRIS exploits temporality and the observation of intervention targets to identify scalar *and* multidimensional causal factors. Furthermore, by introducing a normalizing flow, we extend CITRIS to leverage and disentangle representations obtained by already pretrained autoencoders. Extending previous results on scalar causal factors, we prove identifiability in a more general setting, in which only some components of a causal factor are affected by interventions. In experiments on 3D rendered image sequences, CITRIS outperforms previous methods on recovering the underlying causal variables, and can even generalize to unseen instantiations of causal factors, opening future research areas in sim-to-real generalization.

## 1 Introduction

Causal representation learning (Khemakhem et al., 2020; Lachapelle et al., 2021; Locatello et al., 2020; Schölkopf et al., 2021) focuses on learning representations of causal factors from high-dimensional observations, such as images. Commonly, these causal factors are assumed to be scalars. As settings become more complex and high-dimensional, however, so do the causal dynamics, where estimating every scalar causal variable becomes impractical. Consider for instance a set of objects interacting in a three-dimensional space. For each object, we can describe its position by three variables $x, y, z$ and its rotation in multiple angles. However, it is often more natural and sufficient to consider those factors as two *multidimensional* variables, *i.e.* position and rotation, especially when the definition of the default axes is ambiguous. Such multidimensional causal factors are reminiscent of macrovariables explored before in causality (Chalupka et al., 2015; 2016a;b; Hoel et al., 2013; Höltgen, 2021), but they remain yet unexplored in the causal representation learning setup.

In this paper, we consider causal factors as potentially multidimensional vectors. To identify these multidimensional causal variables, we use sequences of observations where interventions may have been performed at any time step. This setup resembles a reinforcement learning environment, with an agent performing actions over time representing interventions. We assume that we can observe the intervention targets in our data, but not the intervention values. We refer to this setup as TempoRal Intervened Sequences (TRIS). While recent works, Lachapelle et al. (2021) and Yao et al. (2021), have considered a similar temporal setup, they do not exploit the knowledge of the intervention targets, and require scalar variables while we generalize to multidimensional causal factors.

In TRIS, we prove that we can identify the *minimal causal variables*, which model solely the information of a causal factor that is strictly effected by a provided intervention. Meanwhile, all information that cannot be directly influenced by interventions is collected in a separate group of latent variables. As a practical implementation of this, we propose CITRIS for Causal Identifiability from TempoRal Intervened Sequences. CITRIS is a variational autoencoder that learns an assignment of latent variables to causal factors, and promotes disentanglement by conditioning each latent's prior distribution only on its respective intervention target. In experiments on 3D rendered video datasets, CITRIS disentangles the causal factors with high accuracy. Moreover, we extend CITRIS to pretrained

---

[*]Qualcomm AI Research is an initiative of Qualcomm Technologies, Inc.

autoencoders. By using a normalizing flow (Rezende & Mohamed, 2015), CITRIS learns a mapping from the entangled autoencoder representation to a disentangled causal representation. We empirically show that the normalizing flow can even generalize its disentanglement to unseen instantiations of causal factors, holding promise for future work on generalization of causal representations.

## 2 IDENTIFIABILITY OF MINIMAL CAUSAL VARIABLES

We first describe our setting, TempoRal Intervened Sequences (TRIS), in which identifying the underlying causal factors is not always possible. Therefore, we define the concept of *minimal causal variables*, which represent the manipulable part of each multidimensional causal factor. Finally, we show under which conditions we can recover the minimal causal variables in TRIS.

### 2.1 TEMPORAL INTERVENED SEQUENCES (TRIS)

In TRIS, we consider data generated by an underlying latent temporal causal process. We assume this process to be a dynamic Bayesian network (DBN) (Dean & Kanazawa, 1989; Murphy, 2002) $G$ over the $K$ causal variables $(C_1, C_2, ..., C_K)$, which is first-order Markov, stationary, and without instantaneous effects. This means that each causal factor $C_i$ is instantiated at each time step $t$, denoted by $C_i^t$, and its causal parents can only be causal factors at time $t-1$, denoted as $C_j^{t-1}$, including its own previous value $C_i^{t-1}$. As opposed to most work on causal representation learning, we allow the causal factor to be potentially multidimensional, *i.e.*, $C_i \in \mathcal{D}_i^{M_i}$ with $M_i \geq 1$ with $\mathcal{D}_i$ being *e.g.* $\mathbb{R}$ for continuous variables. The causal factor space is then defined as $\mathcal{C} = \mathcal{D}_1^{M_1} \times \mathcal{D}_2^{M_2} \times ... \times \mathcal{D}_K^{M_K}$. At each time step $t$, we obtain a high-dimensional observation $X^T$ representing a noisy, entangled view of all causal factors. We define the observation function $h(C_1^t, C_2^t, ..., C_K^t, E_o^t) = X^t$, where $E_o^t$ represents noise independent of the causal factors, and $h : \mathcal{C} \times \mathcal{E} \to \mathcal{X}$ is a bijective function from the causal factor space $\mathcal{C}$ and the space of the noise variables $\mathcal{E}$ to the observation space $\mathcal{X}$. Finally, we assume that in each time-step some causal factors might (or might not) have been intervened upon and that we have access to the corresponding intervention targets, but not the intervention values. We denote these intervention targets by the binary vector $I^t \in \{0, 1\}^K$ where $I_i^t = 1$ refers to an intervention on the causal variable $C_i^t$.

### 2.2 MINIMAL CAUSAL VARIABLES

In TRIS, we generally cannot disentangle two causal factors if they are always intervened upon jointly, or, on the contrary, if they are never intervened upon. A common example for this are two variables, $x$ and $y$, that follow a Gaussian distribution over time. Then, any two orthogonal axes can describe the distribution equally well (Hyvärinen et al., 2001; 2019), making it impossible to uniquely identify them without individual interventions.

Additionally, we cannot even completely reconstruct the multidimensional latent causal factors in TRIS, when by the nature of the system the provided interventions leave some of the causal factor's dimensions unaffected. We formalize this as follows. Suppose for each causal factor $C_i \in \mathcal{D}^{M_i}$, there exists an invertible map $s_i : \mathcal{D}_i^{M_i} \to \mathcal{D}_i^{\text{var}} \times \mathcal{D}_i^{\text{inv}}$ that splits the domain $\mathcal{D}^{M_i}$ of $C_i$ into a part that is variant and a part that is invariant under intervention. We denote the two parts of this map as

$$s_i(C_i^t) = (s_i^{\text{var}}(C_i^t), s_i^{\text{inv}}(C_i^t)) \tag{1}$$

The split $s$ must be invertible, so that we can map back and forth between $\mathcal{D}_i^{M_i}$ and $\mathcal{D}_i^{\text{var}} \times \mathcal{D}_i^{\text{var}}$ without losing information. Furthermore, $s_i^{\text{inv}}(C_i^t)$ must be independent of the intervention, *i.e.* $s_i^{\text{inv}}(C_i^t) \perp\!\!\!\perp I_i^t \mid \text{pa}(C_i^t)$, and both parts of the split must be conditionally independent, *i.e.* $s_i^{\text{inv}}(C_i^t) \perp\!\!\!\perp s_i^{\text{var}}(C_i^t) \mid \text{pa}(C_i^t), I_i^t$. As a result, $s_i^{\text{var}}(C_i^t)$ will contain the manipulable, or *variable*, part of $C_i^t$. In contrast, $s_i^{\text{inv}}(C_i^t)$ is the *invariable* part of $C_i^t$ which is independent of the intervention.

For any causal variable, there exist at least the trivial split where $\mathcal{D}_i^{\text{var}} = \mathcal{D}_i^{M_i}, \mathcal{D}_i^{\text{inv}} = \{0\}$ (no invariant information). But not all splits are trivial, e.g. when some dimensions are invariant. Intuitively, we want to identify the split where $s_i^{\text{var}}$ contains *solely* the manipulable information:

**Definition 2.1.** *The* minimal causal split *of a variable $C_i^t$ with respect to its intervention variable $I_i^t$ is the split $s_i$ which maximizes the information content $H(s_i^{\text{inv}}(C_i^t)|pa(C_i^t))$. Under this split, $s_i^{\text{var}}(C_i^t)$ is defined as the* minimal causal variable.

Here, $H$ denotes the entropy in the discrete case, and the limiting density of discrete points (LDDP) (Jaynes, 1957; 1968) for continuous variables. Intuitively, this ensures that only the information which truly depends on the intervention is represented in $s_i^{\text{var}}(C_i)$. Our goal becomes to identify these minimal causal variables, which depends on the characteristics of the provided intervention.

## 2.3 Learning minimal causal variables

As a practical example of TRIS, we consider a dataset $\mathcal{D}$ of tuples $\{x^t, x^{t+1}, I^{t+1}\}$ where $x^t, x^{t+1} \in \mathbb{R}^N$ represent the observations at time step $t$ and $t+1$ respectively. To learn a causal representation, we consider a latent space $\mathcal{Z}$ larger than the latent causal factor space $\mathcal{C}$, i.e. $\mathcal{Z} \subseteq \mathbb{R}^M, M \geq \dim(\mathcal{E}) + \dim(\mathcal{C})$. In this latent space, we aim to disentangle the causal factors.

Our goal is to approximate the inverse of the observation function $h$ by learning two components. First, we learn an invertible mapping from observations to latent space, $g_\theta : \mathcal{X} \to \mathcal{Z}$. Second, we learn an assignment function $\psi : [\![1..M]\!] \to [\![0..K]\!]$ that maps each dimension of $\mathcal{Z}$ to a causal factor. Learning a flexible assignment function $\psi$ allows us to allocate any dimension size per causal factor without knowing the individual dimensions $M_1, ..., M_K$ in advance. Further, some variables like circular angles or categorical factors with many categories can have simpler distributions when modelled in more dimensions. In addition to the $K$ causal factors, we use $\psi(j) = 0, j \in [\![1..M]\!]$ to indicate that the latent dimension $z_j$ does not belong to any minimal causal variable. Instead, those dimensions might model $s_i^{\text{inv}}(C_i)$ for some causal factor $C_i$ or the observation noise $E_o^t$. Finally, we denote the set of latent variables that $\psi$ assigns to the causal factor $C_i$ with $z_{\psi_i} = \{z_j | j \in [\![1..M]\!], \psi(j) = i\}$.

To enforce a disentanglement of causal factors, we model a prior distribution in latent space, $p_\phi(z^{t+1}|z^t, I^{t+1})$, with $z^t, z^{t+1} \in \mathcal{Z}$, $z^t = g_\theta(x^t), z^{t+1} = g_\theta(x^{t+1})$. This transition prior enforces a disentanglement by conditioning each latent variable on exactly one of the intervention targets:

$$p_\phi\left(z^{t+1}|z^t, I^{t+1}\right) = \prod_{i=0}^{K} p_\phi\left(z_{\psi_i}^{t+1}|z^t, I_i^{t+1}\right) \tag{2}$$

where $I_0^{t+1} = 0$. Then, the objective of the model is to maximize the likelihood $p_{\phi,\theta}(x^{t+1}|x^t, I^{t+1})$. Under the assumptions in Section 2.1, we can prove the following identifiability result for this setup:

**Theorem 2.2.** *Suppose that $\phi^*, \theta^*$ and $\psi^*$ are the parameters that, under the constraint of maximizing the likelihood $p_{\phi,\theta}(x^{t+1}|x^t, I^{t+1})$, maximize the information content of $p_\phi(z_{\psi_0}^{t+1}|z^t)$. Then, the model $\phi^*, \theta^*, \psi^*$ learns a latent structure where $z_{\psi_i}^{t+1}$ models the minimal causal variable of $C_i$ if $C_i^{t+1} \not\perp\!\!\!\perp I_i^{t+1}|C^t, I_j^{t+1}$ for any $i \neq j$. All remaining information is modeled in $z_{\Psi_0}$.*

We provide the proof for this statement in Appendix A. Finding the minimal variables intuitively means that the latent variables $z_{\psi_i}$ model only the information of $C_i$ which strictly depends on the intervention target $I_i^{t+1}$, thus defining causal variables by their intervention dependency.

# 3 Causal Identifiability from Temporal Intervened Sequences

To identify causal factors from temporal observations with interventions, we propose CITRIS (Causal Identifiability from Temporal Intervened Sequences). Below, we discuss its architecture and variants.

## 3.1 Variational Autoencoder setup

Inspired by previous works (Locatello et al., 2020; Träuble et al., 2021), we implement the framework of Section 2.3 by learning a variational autoencoder (VAE) (Kingma & Welling, 2014), visualized in Figure 1. The encoder $q_\theta$ and decoder $p_\theta$ approximate the invertible mapping $g_\theta$ from observations to latent space, and $p_\phi\left(z^{t+1}|z^t, I^{t+1}\right)$ is the transition prior on the latent variables. In this VAE setup, the objective of the model becomes the Evidence Lower Bound (ELBO):

$$\mathcal{L}_{\text{ELBO}} = -\mathop{\mathbb{E}}_{z^{t+1}}\left[\log p_\theta\left(x^{t+1}|z^{t+1}\right)\right] + \mathop{\mathbb{E}}_{z^t, \psi}\left[\sum_{i=0}^{K} D_{\text{KL}}\left(q_\theta(z_{\psi_i}^{t+1}|x^{t+1})||p_\phi(z_{\psi_i}^{t+1}|z^t, I_i^{t+1})\right)\right] \tag{3}$$

The KL divergence uses the prior definition of Equation (2). This ensures that, conditioned on the previous time step and the interventions, the different blocks of latent variables are independent.

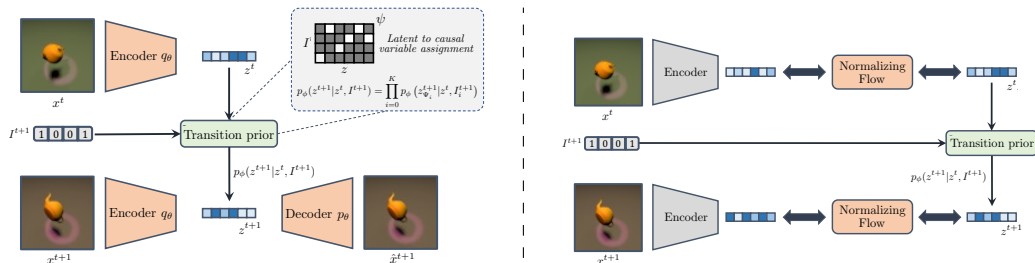

Figure 1: Comparing the VAE and AE+NF setup of CITRIS. **Left**: The transition prior promotes disentanglement in the latent space by conditioning each latent variable on only one intervention target. **Right**: The Normalizing Flow learns to map the autoencoder latents to a new, disentangled space.

Thereby, the assignment function of latent to causal variables, $\psi$, is learned via a Gumbel-Softmax distribution (Jang et al., 2017) per latent variable. Hence, during training, we sample a latent-to-causal variable assignment from these distributions, while for inference, we can use the argmax to obtain a unique assignment. To encourage information independent of any intervention to be modeled in $z_{\psi_0}$, we weight the KL divergence of $z_{\psi_0}$ with $1 - \lambda$, where $\lambda > 0$ is a hyperparameter (usually $\lambda = 0.01$).

The prior $p_\phi$ for each set of latents $z_{\psi_i}^{t+1}$ is implemented by an autoregressive model. For each set of latents $z_{\psi_i}^{t+1}$, the model takes $z^t$, $I_i^{t+1}$ and $z^{t+1}$ as input, where we sample from $\psi$ and mask the dimensions of $z^{t+1}$ for which $\psi(j) \neq i$. From this input, the model predicts one Gaussian distribution per latent variable. The autoregressive nature of the prior allows complex distributions over the multiple latent dimensions, while still being independent across causal variables.

## 3.2 Using Pretrained Autoencoders

In practice, VAEs can struggle to model high-dimensional complex images, especially when small details in the image are relevant. To overcome this issue, we propose an adaptation of CITRIS to pretrained autoencoders. In this setting, the invertible map $g_\theta$ is implemented by a deep autoencoder, which is trained on observational data without interventions independently of any disentanglement. In a second step, we freeze the autoencoder and learn a normalizing flow (Rezende & Mohamed, 2015) that maps the entangled latent representation to a disentangled version. The invertibility of the normalizing flow ensures that no information is lost when mapping from the entangled to disentangled space, and thus we can use the pretrained decoder to reconstruct the observations without requiring any fine-tuning. Compared to the VAE setup, we replace the encoder by a successive application of the frozen encoder and a normalizing flow, shown in Figure 1. Besides that, we deploy the same setup of the transition prior structure. This setup provides an opportunity for generalizing causal factors beyond the known dataset, and we verify the viability of this approach in a restricted setting in Section 4.2.

## 4 Experiments

### 4.1 Experimental setup

**Temporal Causal3DIdent** We evaluate CITRIS on an adapted, temporal version of the Causal3DIdent identifiability benchmark (von Kügelgen et al., 2021). The dataset consists of 3D renderings of objects with seven different causal factors: the object position as 3D vector; the object rotation as 2D vector; the hue of the object, background and spotlight; the spotlight's rotation; and the object shape. The relations among those variables are inspired by the setup of von Kügelgen et al. (2021) and include various common causal structures like confounders and chains. Each variable follows a Gaussian distribution over time, where the mean is a (non-linear) function of the parents. We perform perfect interventions with $I_i^t \sim \text{Bernoulli}(0.1)$, ensuring the minimal causal variables to be the true factors.

**Baselines** We compare CITRIS to SlowVAE (Klindt et al., 2021). Notably, SlowVAE assumes that the factors of variation are independent, which often cannot be met in more complex settings like Temporal Causal3DIdent, as shown in the experiments. The second baseline is iVAE (Khemakhem et al., 2020), which we condition on the previous time step observation $x^t$ and the intervention targets

$I^{t+1}$. As we aim to find a mapping from image to a causal space, which is independent of those factors, we must adapt iVAE to only condition its prior on $u$. We refer to this model variant as iVAE$^*$.

**Correlation metrics** Following common practice, we report the correlation of the learned latent variables to the ground truth causal factors. Since in our setup, multiple latent variables can jointly describe a single causal variable, we first learn a mapping between a set of latents and all causal variables, *e.g.*, with an MLP. During this learning process, no gradients are propagated through the model. The baselines, iVAE$^*$ and SlowVAE, do not learn an assignment of latent to causal factors. Instead, we assign each latent dimension to the causal factor it has the highest correlation with, which gives the baselines a considerable advantage over CITRIS. We report both the $R^2$ coefficient of determination (Wright, 1921) and the Spearman's rank correlation coefficient (Spearman, 1904). To better spot spurious correlations between latents and causal factors, we measure the correlations on a test dataset for which we sample the causal factors independently.

### 4.2 Temporal Causal3DIdent experiments

**7-shapes experiments** We apply all models to the Temporal Causal3DIdent dataset with all 7 object shapes (see Table 1). Both the VAE and Normalizing Flow version of CITRIS considerably outperform the two baselines and are able to achieve an average $R^2$ and Spearman correlation above 0.9, while keeping the correlation between factors low. Moreover, CITRIS-NF achieves close-to optimal scores, and especially outperforms the VAE-based approaches in modeling the rotations. This underlines the optimization benefits of using pretrained autoencoders for disentanglement learning on complex, high-dimensional observations. In contrast, the SlowVAE entangles the causal factors due to their strong correlation over time. Meanwhile, in the iVAE$^*$, the hue of the spotlight was highly entangled since its appearance differs with different background and object colors.

Table 1: Results on the Temporal-Causal3DIdent dataset. *diag* refers to the average score of the predicted causal factor to its true value (optimal 1), and *sep* the maximum correlation per predicted causal variable besides its true factor (optimal 0). CITRIS disentangles the causal factors well, with CITRIS-NF achieving close-to optimal scores.

| Methods | $R^2$ diag ↑ | $R^2$ sep ↓ | Spearman diag ↑ | Spearman sep ↓ |
|---|---|---|---|---|
| SlowVAE | 0.61 | 0.23 | 0.59 | 0.27 |
| iVAE$^*$ | 0.80 | 0.29 | 0.77 | 0.28 |
| CITRIS-VAE | 0.89 | 0.10 | 0.88 | 0.12 |
| CITRIS-NF | **0.98** | **0.04** | **0.97** | **0.08** |

**Generalization of causal representations** Finally, we evaluate whether the causal representations of CITRIS can generalize to new, unseen settings. For this, we reuse the same autoencoder as before, but train the Normalizing Flow on an interventional dataset which excludes any observations from two shapes. Afterwards, we test its zero-shot generalization to the two unseen shapes. Note that optimal performance cannot be achieved here, since the central point and default rotation of an object cannot be generalized to other objects. Nonetheless, the results in Table 2 indicate a strong disentanglement among factors, with slight decreases in position and rotation due to the forementioned limitations. This shows that the learned disentanglement function can indeed generalize to unseen instantiations of causal factors, promising potential for future work on generalizing causal representations to unseen settings with CITRIS.

Table 2: Results on the Temporal-Causal3DIdent dataset with CITRIS-NF trained on 5 object shapes. The same metrics as in Table 1 are reported. Even for 2 unseen shapes, CITRIS-NF can disentangle their causal factors well.

| CITRIS-NF | $R^2$ diag ↑ | $R^2$ sep ↓ | Spr. diag ↑ | Spr. sep ↓ |
|---|---|---|---|---|
| 5 seen shapes | 0.98 | 0.05 | 0.97 | 0.10 |
| 2 unseen shapes | 0.94 | 0.15 | 0.93 | 0.19 |

## 5 Conclusion

We propose CITRIS, a VAE framework for learning causal representations. CITRIS identifies the minimal causal variables of a dynamical system from temporal, intervened sequences. Furthermore, by using normalizing flows, CITRIS learns to disentangle the representation of pretrained autoencoders. In experiments, CITRIS reliably recovered the causal factors of 3D rendered images. Moreover, we empirically showed that CITRIS can generalize to unseen instantiations of causal factors. This promises great potential for future work on simulation-to-real generalization research for causal representation learning. As future work, CITRIS can be extended to an active learning setup, allowing for more data-efficient causal identifiability methods in practice.

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

# Appendix

## TABLE OF CONTENTS

## A   PROOFS

The following section contains the proof for Theorem 2.2. We first give an overview of the notation and additional preliminary discussions in Appendix A.1. Then, we give an outline of the proof. The remaining sections provide the details of the proof.

### A.1   PRELIMINARIES

#### A.1.1   SUMMARY OF NOTATION

We summarize the notation, which is in most cases the same as used for the main paper and extend it in small aspects for the proof, as follows:

- We assume $K$ causal factors $C_1, \ldots, C_K$ such that $C_i \in \mathcal{D}_i^{M_i}$ with $M_i \geq 1$;
- We can group all causal factors in a single variable $C = (C_1, \ldots, C_K) \in \mathcal{C}$, where $\mathcal{C}$ is the causal factor space $\mathcal{C} = \mathcal{D}_1^{M_1} \times \mathcal{D}_2^{M_2} \times ... \times \mathcal{D}_K^{M_K}$;
- The data is generated by a latent Dynamic Bayesian network with variables $(C_1^t, C_2^t, ..., C_K^t)_{t=1}^{T}$;
- We assume to know at each time step the binary intervention vector $I^t \in \{0, 1\}^{K+1}$ where $I_i^t = 1$ refers to an intervention on the causal factor $C_i^{t+1}$. As a special case $I_0^t = 0$ for all $t$;
- For each causal factor $C_i$, there exists a split $s_i^{\mathrm{var}}(C_i), s_i^{\mathrm{inv}}(C_i)$ such that $s_i^{\mathrm{var}}(C_i)$ represents the variable/manipulable part of $C_i$, while $s_i^{\mathrm{inv}}(C_i)$ represents the invariable part of $C_i$;
- The minimal causal split is defined as the one which only contains the intervention-dependent information in $s_i^{\mathrm{var}}(C_i)$, and everything else in $s_i^{\mathrm{inv}}(C_i)$. This split is denoted by $s_i^{\mathrm{var}^*}(C_i)$ and $s_i^{\mathrm{inv}^*}(C_i)$
- At each timestep we can access observations $x^t, x^{t+1} \in \mathcal{X} \subseteq \mathbb{R}^N$;
- Observation function $h : \mathcal{C} \times \mathcal{E} \rightarrow \mathcal{X}$, where $\mathcal{E}$ is the space of the noise variables;
- Latent vector $z^t \in \mathcal{Z} \subseteq \mathbb{R}^M$, where $\mathcal{Z}$ is the latent space of dimension $M \geq \dim(\mathcal{E}) + \dim(\mathcal{C})$;
- Inverse of the observation function in the latent space $g_\theta : \mathcal{X} \rightarrow \mathcal{Z}$;
- Assignment function from latent dimensions to causal factors $\psi : [\![1..M]\!] \rightarrow [\![0..K]\!]$;
- Disentanglement function $\delta^* : \mathcal{X} \rightarrow \tilde{\mathcal{C}} \times \tilde{\mathcal{E}}$ with $\tilde{\mathcal{C}} = \mathcal{D}^{\tilde{M}_1} \times ... \times \mathcal{D}^{\tilde{M}_K}$ and $\tilde{M}_i$ being the number of latent dimensions assigned to causal factor $C_i$ by $\psi^*$. We denote the output of $\delta^*$ for an observation $X$ as $\delta^*(X) = (\tilde{C}_1, \tilde{C}_2, ..., \tilde{E})$. Then, $\delta^*$ is a disentanglement function if there exist a set of deterministic functions $h_0, h_1, ..., h_K$ for which, for any $X = h(C, E)$, $h_i(\tilde{C}_i) = C_i$ for all $i \in [\![1..K]\!]$, and $h_0(\tilde{E}) = E$.

- The representation of $\delta^*$ in terms of the learnable function is denoted by $g_\theta^*$ and $\psi^*$;
- Latent variables assigned to each causal factor $C_i$ by $\psi$ are denoted as $z_{\psi_i} = \{z_j | j \in [\![1..M]\!], \psi(j) = i\} = \{g_\theta(x^t)_j | j \in [\![1..M]\!], \psi(j) = i\}$;
- The remaining latent variables that are not assigned to any causal factor are denoted as $z_{\psi_0}$;
- The goal is to learn for each $C_i$: $p_\phi\left(z_{\psi_i}^{t+1}|z^t, I_i^{t+1}\right) \approx p\left(s_i^{\mathrm{var}}(C_i^{t+1})|C^t, I_i^{t+1}\right)$;

### A.1.2 Limiting density of discrete points

In this section, we give a short overview on the difference between differential entropy and the limiting density of discrete points approach, introduced by Jaynes (1957; 1968). Differential entropy on a continuous random variable $X$ with a distribution $p(X)$ is defined as:

$$H(X) = -\int p(X) \log p(X) dx \tag{4}$$

While for discrete variables, entropy has the intuitive explanation of an uncertainty measure, or the 'information' of a variable, one cannot draw the same relation so easily for continuous variables. This is because differential entropy lacks properties that would be necessary for that. For one, the entropy can become negative. Secondly, and most importantly for the use case in this paper, it is not invariant under invertible transformations, *i.e.* a change of variables. For the example of the random variable $X$, the entropy of $H(X)$ does not necessarily equal to $H(aX)$ where $a$ is a constant factor, *e.g.* $a = 2$. Thus, it becomes difficult to use differential entropy as a measure of information content of a continuous variable, like in the discrete case.

One approach that was proposed to overcome these issues is the limiting density of discrete points (LDDP) Jaynes (1957; 1968). It adjusts the definition of differential entropy by introducing an *invariant measure* $m(X)$, which can be seen as a reference distribution we measure the entropy of $p(X)$ to. Intuitively, the LDDP adjustment is derived from arguing that the continuous entropy should be derived by taking the limit of increasingly dense discrete distributions. In the limit of infinitely many discrete points, one arrives at the entropy for continuous functions, which becomes:

$$H(X) = -\int p(X) \log \frac{p(X)}{m(X)} dx \tag{5}$$

Note that in some formulations, a constant $\log N$ is added to this equation, where $N$ is the number of discrete points considered which goes against infinity in the limit. Since for this paper, we only require to compare two entropy values with each other and do not require the entropy to take a specific value, we can neglect this constant.

One crucial property of LDDP, which we use in the following proof, is that the entropy stays invariant under a change of variable. This is achieved by transforming the invariant measure $m(X)$ by the exact same invertible transformation as done for $p(X)$. Therefore, when coming back to the example of scaling $X$ by a constant factor, both $p(X)$ and $m(X)$ change in the same way, resulting in $H(X) = H(aX)$.

### A.2 Proof outline

The goal of this section is to proof Theorem 2.2: the global optimum of CITRIS will find the minimal causal variables. We will take the following steps in the proof:

1. (Appendix A.3) Firstly, we show that the function $\delta^*$ that disentangles the true latent variables $C_1, ..., C_K$ and assigns them to the corresponding sets $z_{\psi_1}, ..., z_{\psi_K}$ constitutes a global, but not necessarily unique, optimum for maximizing the likelihood of Equation (2).
2. (Appendix A.4) Next, we characterize the class of disentanglement functions $\Delta^*$ which all represent a global maximum of the likelihood, *i.e.* get the same score as the true disentanglement. In particular, we show that in all optimal disentanglement functions, each assignment set $z_{\psi_i}$ contains the variable part of the causal factor $s_i^{\mathrm{var}}(C_i)$, but that it might contain also the invariable parts of any other causal factor, thus creating multiple optimal solutions. We do this in two sub-steps:
    (a) First, we assume that all intervention targets are independent, *i.e.* $I_i^{t+1} \perp\!\!\!\perp I_j^{t+1}|C^t$ for any $i \neq j$.

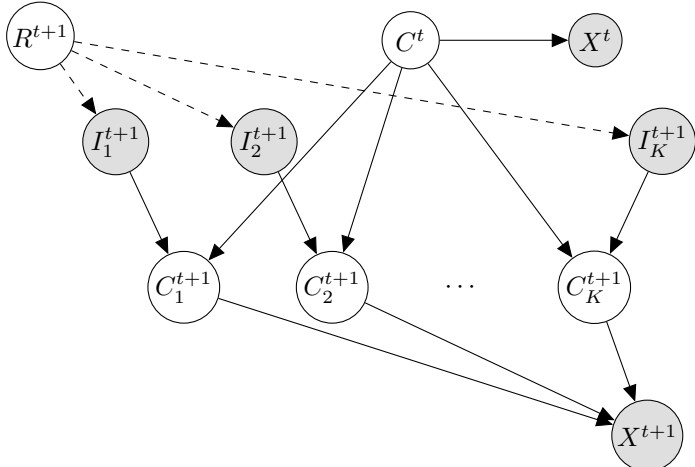

Figure 2: Temporal causal graph with the latent causal factors $C_1^{t+1}, ..., C_K^{t+1}$ and $C^t = \{C_1^t, ..., C_K^t\}$, the observed intervention targets $I_1^{t+1}, ..., I_K^{t+1}$ with a latent confounder $R^{t+1}$ representing the regime, and the observations $X^t$ and $X^{t+1}$. Observed variables are shown in gray, latent variables in white. For simplicity, the variable $C^t$ summarizes all causal factors at time $t$ and inherits all edges of those. The dashed lines from $R^{t+1}$ represent that there can exist an arbitrary confounder between intervention targets, but are not strictly necessary. The goal is to identify the causal factors $C_1, ..., C_K$.

    (b) Secondly, we extend it to a wider group of intervention settings where interventions might be confounded, and show that all of them fall in the same class $\Delta^*$.

3. (Appendix A.5) Finally, we derive Theorem 2.2 by showing that the function $\hat{\delta} \in \Delta^*$, which maximizes the entropy of $z_{\psi_0}$, identifies the minimal causal mechanisms, which intuitively represent the parts of the causal factors that are affected by the available interventions.

Along the way, we will make use of Figure 2 summarizing the temporal causal graph. For the remainder of the proof, we assume that the prior $p_\phi\left(z^{t+1}|z^t, I^{t+1}\right)$ and the invertible map $g_\theta$ are sufficiently complex to approximate any possible function and distribution one might encounter. To simplify the exposition, we also assume that the latent dimension size is unlimited, *i.e.* $M = \infty$, so there are no limitations on how many latent variables $z_{\psi_i}$ can be used to represent a causal factor $C_i$. In practice, however, this is not a limiting factor as long as we can overestimate the dimensions of the causal factors and noise variables.

Throughout the proof, we will use $C^t$ to refer to the set of all causal factors at time step $t$, *i.e.* $C^t = \{C_1^t, ..., C_K^t\}$. Similarly, we define $I^{t+1} = \{I_1^{t+1}, ..., I_K^{t+1}\}$.

### A.3    STEP 1: TRUE DISENTANGLEMENT IS ONE OF THE GLOBAL MAXIMA OF THE CONDITIONAL LIKELIHOOD

We start by proving the following Lemma:

**Lemma A.1.** *The true disentanglement function $\delta^*$ that correctly disentangles the true causal factors $C_1^{t+1}, ..., C_K^{t+1}$ from observations $X^t, X^{t+1}$ using the true $\psi^*$ assignment function on the true latent variables $Z^{t+1}$ is one of the global maxima of the likelihood of $p(X^{t+1}|X^t, I^{t+1})$.*

We are interested in optimizing $p(X^{t+1}|X^t, I^{t+1})$. We can first try to simplify this equation with the knowledge of the causal graph in Figure 2, *i.e.* using the true underlying generative model, since we aim to show that learning the causal factors and aligning them correspondingly in the prior of Equation (2) represents a global optimum of maximizing $p(X^{t+1}|X^t, I^{t+1})$. Using the conditional independence relations of the graph in Figure 2, we write the joint distribution of all the variables in

the true generative model as:

$$p(X^t, X^{t+1}, C^t, C^{t+1}, I^{t+1}) = p(X^{t+1}|C^{t+1}) \cdot \left[ \prod_{i=1}^{K} p(C_i^{t+1}|C^t, I_i^{t+1}) \right] \cdot p(X^t|C^t) \cdot p(C^t) \cdot p(I^{t+1})$$

(6)

We can now condition on $X^t$ and $I^{t+1}$, marginalize out $C^t$ and $C^{t+1}$ and write the conditional likelihood as:

$$p(X^{t+1}|X^t, I^{t+1}) = \int_{C^{t+1}} \int_{C^t} p(X^{t+1}|C^{t+1}) \cdot \left[ \prod_{i=1}^{K} p(C_i^{t+1}|C^t, I_i^{t+1}) \right] \cdot p(C^t|X^t) dC^t dC^{t+1}$$

(7)

In our assumptions of Section 2.1, we have defined the observation function $h$ to be bijective, meaning that there exists an inverse $f$ that can identify the causal factors $C^t$ and noise variable $E_o^t$ from $X^t$. Thus, we can write $p(C^t|X^t) = \delta_{f(X^t)=C^t}$, where $\delta$ is a Dirac delta. Since the noise on the observations, $E_o^t$, is said to be independent of $X^{t+1}$ and $C^{t+1}$, we can remove it from being in the conditioning set. This leads us to:

$$p(X^{t+1}|X^t, I^{t+1}) = \int_{C^{t+1}} \left[ \prod_{i=1}^{K} p(C_i^{t+1}|C^t, I_i^{t+1}) \right] \cdot p(X^{t+1}|C^{t+1}) dC^{t+1}$$

(8)

Since we have assumed $h$ to be bijective, we know that for each $X^{t+1}$, there exist only one combination of $C^{t+1}$ and $E_o^{t+1}$. Thus, by using the change of variables formula, we can rewrite aboves equation by:

$$p(X^{t+1}|X^t, I^{t+1}) = |J_h|^{-1} \cdot \left[ \prod_{i=1}^{K} p(C_i^{t+1}|C^t, I_i^{t+1}) \right] \cdot p(E_o^{t+1})$$

(9)

where $J_h = \frac{\partial h(C^{t+1}, E_o^{t+1})}{\partial C^{t+1} \partial E_o^{t+1}}$ denotes the Jacobian of the bijective/invertible observation function $h$. Equation (9) constitutes a global optimum of the maximum likelihood, since it represents the true underlying dynamics.

We relate this conditional likelihood to the prior setup of CITRIS. We show that assigning $C_i^{t+1}$ to $z_{\psi_i}^{t+1}$, i.e., learning the true assignment function $\psi^*$, provides us with the same maximum likelihood solution as in Equation (9). We have defined our objective in Section 2 in Equation (2) as:

$$p_\phi\left(z^{t+1}|z^t, I^{t+1}\right) = \prod_{i=0}^{K} p_\phi\left(z_{\psi_i}^{t+1}|z^t, I_i^{t+1}\right)$$

(10)

Since we know that $g_\theta^*$ is an invertible function between $\mathcal{X}$ and $\mathcal{Z}$, we know that $z^t$ must include all information of $X^t$. Thus, we can also replace it with $z^t = [C^t, E_o^t]$, giving us:

$$p_\phi\left(z^{t+1}|C^t, E_o^t, I^{t+1}\right) = \prod_{i=0}^{K} p_\phi\left(z_{\psi_i}^{t+1}|C^t, E_o^t, I_i^{t+1}\right)$$

(11)

The optimal assignment function $\psi^*$ assigns sufficient dimensions to each causal factor $C_1, ..., C_K$. Since $\mathcal{Z}$ can have a larger space than $\mathcal{E} \times \mathcal{C}$, but $\mathcal{E} \times \mathcal{C}$ is sufficient to describe $\mathcal{X}$, we know that the remaining dimensions of $\mathcal{Z}$ do not contain any information. Thus, the assignment function $\psi^*$ can map them to any causal factor without a change in distribution. Using this assignment function, we now consider $z_{\psi_i^*}^{t+1} = C_i^{t+1}$ for $i = 1, ..., K$. Then, Equation (11) becomes:

$$p_\phi\left(z^{t+1}|C^t, E_o^t, I^{t+1}\right) = \left[ \prod_{i=1}^{K} p_\phi\left(z_{\psi_i^*}^{t+1} = C_i^{t+1}|C^t, I_i^{t+1}\right) \right] \cdot p(z_{\psi_0^*}^{t+1}|C^t, E_o^t)$$

(12)

where we remove $E_o^t$ from the conditioning set for the causal factors, since know that $C^{t+1}$ and $E_o^{t+1}$ is independent of $E_o^t$. We further simplify by noting that $z_{\psi_0^*}^{t+1} = E_o^{t+1}$ is independent of any other factor.

$$p_\phi\left(z^{t+1}|C^t, E_o^t, I^{t+1}\right) = \left[ \prod_{i=1}^{K} p_\phi\left(z_{\psi_i^*}^{t+1} = C_i^{t+1}|C^t, I_i^{t+1}\right) \right] \cdot p(z_{\psi_0^*}^{t+1} = E_o^{t+1})$$

(13)

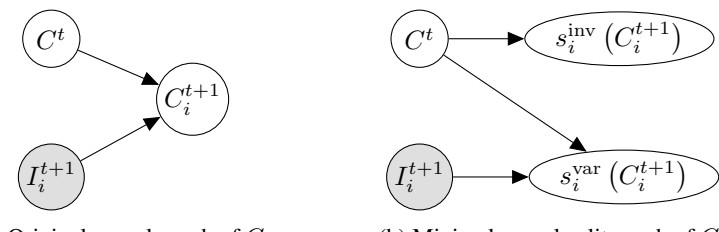

(a) Original causal graph of $C_i$          (b) Minimal causal split graph of $C_i$

Figure 3: Splitting the causal variable $C_i$ in its minimal causal split. (a) In the original causal graph, $C_i^{t+1}$ has $C^t$ (or an eventual subset of it) and $I_i^{t+1}$ as its parents. (b) In the minimal causal split, only the variable part $s_i^{\text{var}}(C_i^{t+1})$ depends on the intervention. The invariable part, $s_i^{\text{inv}}(C_i^{t+1})$, is independent of $I_i^{t+1}$, hence giving us an additional conditional independence. Note that $s_i^{\text{var}}(C_i^{t+1})$ and $s_i^{\text{inv}}(C_i^{t+1})$ are conditionally independent.

Finally, by using $g_\theta^*$, we can replace the distribution on $z^{t+1}$ by a distribution on $X^{t+1}$ by the change of variables formula:

$$
p_\phi\left(X^{t+1}|C^t, E_o^t, I^{t+1}\right) = \left|\frac{\partial g_\theta^*(z^{t+1})}{\partial z^{t+1}}\right| \cdot \left[\prod_{i=1}^K p_\phi\left(z_{\psi_i^*}^{t+1} = C_i^{t+1}|C^t, I_i^{t+1}\right)\right] \cdot p(z_{\psi_0^*}^{t+1} = E_o^{t+1})
\tag{14}
$$

Thereby, it is apparent that $g_\theta^*$ is equal to $h^{-1}$, since both are identical invertible functions between the same spaces ($\mathcal{Z}$ becomes $\mathcal{E} \times \mathcal{C}$ here). Hence, Equation (14) represents the exact same distribution as Equation (9). Therefore, we have shown that the function $\delta^*$ that disentangles the true latent variables $C_1, ..., C_K$ and assigns them to the corresponding sets $z_{\psi_1}, ..., z_{\psi_K}$ constitutes a global, but not necessarily unique, optimum for maximizing the likelihood of Equation (2).

Note that while assigning $C_i^{t+1}$ to $z_{\psi_i}^{t+1}$ provides us with the same maximum likelihood solution as in Equation (9), this is not the the only possible representation. Additional possible representation will be discussed in Step 2.

## A.4    Step 2: Characterizing the Disentanglement Class

Showing that the correct disentanglement constitutes a global optimum is not sufficient for showing that a model trained on solving the maximum likelihood solution converges to it, since there might potentially be multiple global optima. Hence, this section discusses the class of causal representation functions $\delta \in \Delta^*$ which can achieve the same maximum likelihood optimum as the true causal factor disentanglement discussed in Appendix A.3. For this, we first need to distinguish between the *variable* and *invariable* information of a causal variable $C_i$, which is introduced in Appendix A.4.1. Next, we will discuss the causal representation function class $\Delta$ for the setting where interventions are independent, *i.e.* $I_i^{t+1} \perp\!\!\!\perp I_j^{t+1}|C^t$ for any $i \neq j$, and finally extend it to confounded interventions.

### A.4.1    Intervention-independent Variables

Interventions allow us to identify a causal variable by seeing the caused change in its conditional distribution. However, especially when talking about multidimensional causal variables, one might have interventions that only affect a subset of the actual causal variable dynamics, while the rest remains independent of the intervention. As we will see later, this can have an influence on the identifiability result, making the found causal factors intervention-dependent.

We start by considering a single causal factor $C_i \in \mathcal{D}_i^{M_i}$ in the setup of Figure 2 under our previously discussed assumptions. Suppose for each causal factor $C_i \in \mathcal{D}^{M_i}$, there exists an invertible map $s_i : \mathcal{D}_i^{M_i} \to \mathcal{D}_i^{\text{var}} \times \mathcal{D}_i^{\text{inv}}$ that splits the domain $\mathcal{D}^{M_i}$ of $C_i$ into a part that is invariant and a part that is variant under intervention. We denote the two parts of this map as

$$
s_i(C_i^t) = (s_i^{\text{var}}(C_i^t), s_i^{\text{inv}}(C_i^t))
\tag{15}
$$

The split $s$ must be invertible, so that we can map back and forth between $\mathcal{D}_i^{M_i}$ and $\mathcal{D}_i^{\text{var}} \times \mathcal{D}_i^{\text{var}}$

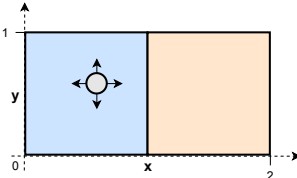

Figure 4: Example for splitting a causal variable into an intervention-dependent and -independent part. The two ground truth causal variables are the $x$ and $y$ positions of the ball. See Appendix A.4.1 for details.

without losing information. Furthermore, to be called a split, $s$ must satisfy $s_i^{\mathrm{inv}}(C_i^t) \perp\!\!\!\perp I_i^t \mid \mathrm{pa}(C_i^t)$, *i.e.*, $s_i^{\mathrm{inv}}(C_i^t)$ is independent of the intervention variable $I_i^t$ given the parents of $C_i^t$. Further, both parts of the split must be conditionally independent, *i.e.* $s_i^{\mathrm{inv}}(C_i^t) \perp\!\!\!\perp s_i^{\mathrm{var}}(C_i^t) \mid \mathrm{pa}(C_i^t), I_i^t$. Hence, we can write their distributions as:

$$p\left(s_i(C_i^{t+1})|C^t, I_i^{t+1}\right) = p\left(s_i^{\mathrm{var}}(C_i^{t+1})|C^t, I_i^{t+1}\right) \cdot p\left(s_i^{\mathrm{inv}}(C_i^{t+1})|C^t\right) \tag{16}$$

This means that $s_i^{\mathrm{var}}(C_i^t)$ will contain the manipulable, or *variable*, part of $C_i^t$. In contrast, $s_i^{\mathrm{inv}}(C_i^t)$ is the *invariable* part of $C_i^t$ which is independent of the intervention. This relation is visualized in Figure 3.

For any causal variable, there may exist multiple possible splits, but there is always at least the trivial split where $\mathcal{D}_i^{\mathrm{var}} = \mathcal{D}_i^{M_i}$ is the original domain of $C_i$, and $\mathcal{D}_i^{\mathrm{inv}} = \{0\}$ is the one-element set (no invariant information). However, there might also exist splits in which $s_i^{\mathrm{inv}}(C_i^{t+1}) \neq \emptyset$. For instance, in a multidimensional causal variable $\hat{C} \in \mathbb{R}^3$, if an intervention only affects the first two dimensions while the last one remains unaffected, we obtain the split $s^{\mathrm{var}}([\hat{C}_1, \hat{C}_2]), s^{\mathrm{inv}}(\hat{C}_3)$. Nonetheless, this can even happen for scalar variables, since we do not constraint the possible distributions of $C_i$. We give an example for such a case below.

**Example 1** Consider the scenario in Figure 4 where we have a ball with its two positional dimensions $x$ and $y$ as its causal factors. For now, we only focus on its $x$ position (in the remainder of the section, $x^t$ refers to the position of the ball on the $x$-axis, not the full observation $X^t$ which we denote by a capital letter). Over time, the ball moves within one of the two boxes, but cannot jump in between boxes. An example of such a conditional could be:

$$p(x^{t+1}|x^t, I_x^{t+1} = 0) = \begin{cases} \min(\max(x^t + \epsilon, 0), 1) & \text{if } x^t < 1 \\ \min(\max(x^t + \epsilon, 1), 2) & \text{otherwise} \end{cases} \tag{17}$$

with $\epsilon \sim \mathcal{N}(0, 0.1)$. Intuitively, the ball therefore moves randomly around its previous position, while being bounded by the box it is in. Due to its modular conditional distribution, we can rewrite the causal variable $x$ and its distribution in terms of two different variables: its position within its current box, $u \in [0, 1]$, and a binary variable indicating in which box the ball is, $b \in 0, 1$ (left/blue vs right/orange in Figure 4). Then, its conditional distribution becomes:

$$p(x^{t+1}|x^t, I_x^t) = p\left(b^{t+1}|x^t, I_x^{t+1}\right) \cdot p\left(u^{t+1}|x^t, I_x^{t+1}\right) \tag{18}$$

Now, suppose that an intervention $I_x^{t+1}$ changes the box the ball is in, while the relative position keeps evolving as it would under no intervention, *i.e.* still depending on its parents. Then, we can yet write its conditional distribution as:

$$p(x^{t+1}|x^t, I_x^t) = p\left(b^{t+1}|x^t, I_x^{t+1}\right) \cdot p\left(u^{t+1}|x^t\right) \tag{19}$$

Using the notation above, we therefore can define the split $s^{\mathrm{var}}(x) = b$, $s^{\mathrm{inv}}(x) = u$, where $b$ depends on the intervention, while $u$ does not. Note that $s^{\mathrm{var}}(x) = x$, $s^{\mathrm{inv}}(x) = \emptyset$ is yet another valid split in this case.

**Example 2** Consider the same example as before, however, now with a different intervention setup. Suppose that an intervention $I_x^{t+1}$ constitutes a perfect intervention on $x$, under which $x^{t+1} \perp\!\!\!\perp x^t|I_x^{t+1} = 1$. Then, the previous split $s^{\mathrm{var}}(x) = b$, $s^{\mathrm{inv}}(x) = u$ is not valid anymore, if the intervention target $I_x^{t+1}$ cannot be deterministically deduced from $x^t$, since an intervention changes

the distribution of the relative position $u^{t+1}$. Hence, the only valid split is $s^{\mathrm{var}}(x) = x, s^{\mathrm{inv}}(x) = \emptyset$. This shows that the possible space of such splits depends on the available interventions.

**Minimal causal variables** In the examples above, one can see that for certain situations, a causal variable can have multiple valid splits $s_i^{\mathrm{var}}(C_i), s_i^{\mathrm{inv}}(C_i)$ since intervention-independent information can be modeled in either $s_i^{\mathrm{var}}(C_i)$ or $s_i^{\mathrm{inv}}(C_i)$. The split that will be the most relevant for the identifiability discussion here is the one that assigns only the intervention-dependent information to $s_i^{\mathrm{var}}(C_i)$, and the rest to $s_i^{\mathrm{inv}}(C_i)$. We define this as follows:

**Definition A.2.** *The* minimal causal split *of a variable $C_i^t$ with respect to its intervention variable $I_i^t$ is the split $s_i$ which maximizes the entropy of $H(s_i^{\mathrm{inv}}(C_i^t)|pa(C_i^t))$. Under this split, $s_i^{\mathrm{var}}(C_i^t)$ is defined as the* minimal causal variable *and denoted by $s_i^{\mathrm{var}^*}(C_i^t)$.*

Additionally, we also define:

**Definition A.3.** *The minimal causal mechanism of a variable $C_i$ with respect to its intervention $I_i$ is defined as the conditional distribution $p\left(s_i^{\mathrm{var}^*}(C_i^t)|pa(C_i^{t+1}), I_i^{t+1}\right)$.*

We refer to $p\left(s_i^{\mathrm{var}^*}(C_i^t)|C^t, I_i^{t+1}\right)$ as *minimal* causal mechanism, since it is the distribution for which as little as possible information depends on $I_i^{t+1}$. Hence, the definition of this mechanism depends on the characteristics of the provided intervention. As we will see later, while we cannot guarantee to find the full causal mechanism, we can yet identify the minimal causal mechanism.

The existence of a split where $s^{\mathrm{inv}}(C_i) \neq \emptyset$ for any causal factor $C_i$ creates additional, possible solutions that obtain the same maximum likelihood as the true split. This is because $s^{\mathrm{inv}}(C_i)$ is independent of the intervention target $I_i$, allowing it to be modeled by any set $z_{\psi_j}$ without losing information. The following subsections further characterize the space of new solutions with the existence of such splits.

### A.4.2 INDEPENDENT INTERVENTIONS

In this section, we show by which class of disentanglement functions $\Delta$ a maximum likelihood solution of the generative model can be found.

However, to simplify the first steps, we assume that all intervention targets are independent of each other given the causal factors of the previous time step, *i.e.* $I_i^{t+1} \perp\!\!\!\perp I_j^{t+1}|C^t$ for any $i \neq j \in 1, \ldots, K$. By construction, we also assume $I_0^{t+1} = 0$ for all $t$. We will extend it afterwards in Appendix A.4.3.

**Solutions for $s_i^{\mathrm{inv}}(C_i) = \emptyset$ for all $i \in [\![1..K]\!]$** As a first step, we assume that for all causal factors $C_1, ..., C_K$, there does not exist any minimal causal mechanism split besides $s_i^{\mathrm{inv}}(C_i)$ being the empty set. Therefore, all of $C_i$ is dependent on $I_i^{t+1}$. For this case, consider an arbitrary partition of $C_i$, $s^0(C_i), s^1(C_i)$, with the same invertibility constraints as $s^{\mathrm{var}}, s^{\mathrm{inv}}$ and conditionally independence between $s^0(C_i), s^1(C_i)$, but with a non-empty invariable part, *i.e.*, $s^1(C_i) \neq \emptyset$. For this partition, the conditional entropy of $s^1(C_i)$ given $C^t$ must be strictly lower when conditioning also on $I_i^{t+1}$:

$$H\left(s^1(C_i^{t+1})|C^t, I_i^{t+1}\right) < H\left(s^1(C_i^{t+1})|C^t\right) \tag{20}$$

If the conditional entropy was equal, then $s^1(C_i^{t+1})$ would be independent of $I_i^{t+1}$ given $C^t$, which is only true for $s_i^{\mathrm{inv}}(C_i)$. Since we assume $s_i^{\mathrm{inv}}(C_i)$ is empty, while $s^1(C_i^{t+1})$ is not, this can never happen. Thus, to model a causal factor $C_i$ where $s_i^{\mathrm{inv}}(C_i) = \emptyset$, a maximum likelihood solution can only be found if all of $s_i^{\mathrm{var}}(C_i)$ is conditioned on $I^{t+1}$. Further, since in this setting $C_j^{t+1} \perp\!\!\!\perp C_i^{t+1}|C^t, I^{t+1}$ for all $i, j \in [\![1..K]\!], i \neq j$, there cannot exist any split across multiple causal factors that violate the entropy inequality above for $I_i^{t+1}$ and $I_j^{t+1}$ while still modeling the true conditional distributions.

Similarly, in this setting under the specified assumptions, the information of $I_i^{t+1}$ cannot be determined by any other target variable $I_j^{t+1}, i \neq j$, since otherwise, the targets would not be independent. Hence, we can write the following entropy inequality for any $i \neq j$:

$$H\left(C_i^{t+1}|C^t, I_i^{t+1}\right) < H\left(C_i^{t+1}|C^t\right) = H\left(C_i^{t+1}|C^t, I_j^{t+1}\right) \tag{21}$$

Therefore, one can only achieve the maximum likelihood solution (*i.e.*, the minimum entropy solution)

if all information of $C_i^{t+1}$ is conditioned on $I_i^{t+1}$ for $i = 1, ..., K$ in addition to $C^t$. This implies that each factor $p_\phi(z_{\psi_i}^{t+1}|z^t, I_i^{t+1})$ described in Equation (2) will have $p(C_i^{t+1}|z^t, I_i^{t+1})$, and therefore $z_{\psi_i}^{t+1} = C_i^{t+1}$ as its maximum likelihood solution.

Nonetheless, this excludes $z_{\psi_0}$, *i.e.* the factors independent of any intervention, in this case being the noise $E_o^{t+1}$. Since it is independent of any interventions, any distribution of $E_o^{t+1}$ across the different causal factor sets $z_{\psi_0}, ..., z_{\psi_K}$ will achieve the same likelihood score, as long as the part of $E_o^{t+1}$ across factors is independent. Hence, in conclusion for this scenario, we can guarantee that $z_{\psi_i}$ will model all information of $C_i$ and no other causal factor $C_j, i \neq j$, but can contain additional information from $E^{t+1}$.

**Solutions with invariable parts** Next, we consider the scenario where there exists a split with $s_i^{\text{inv}}(C_i) \neq \emptyset$ for some causal variables in $i \in [\![1..K]\!]$. For this case, we can write the maximum likelihood solution of Equation (9) as:

$$p(X^{t+1}|X^t, I^{t+1}) = \left| \frac{\partial g_\theta^*(z^{t+1})}{\partial z^{t+1}} \right| \cdot \left[ \prod_{i=1}^{K} p(C_i^{t+1}|I_i^{t+1}, C^t) \right] \cdot p(E^{t+1}) \tag{22}$$

$$= \left| \frac{\partial g_\theta^*(z^{t+1})}{\partial z^{t+1}} \right| \cdot \left[ \prod_{i=1}^{K} p(s_i^{\text{var}}(C_i^{t+1})|I_i^{t+1}, C^t) \right] \cdot \left[ \prod_{i=1}^{K} p(s_i^{\text{inv}}(C_i^{t+1})|C^t) \right] \cdot p(E^{t+1}) \tag{23}$$

This equation shows that one can assign $s_1^{\text{inv}}(C_1), ..., s_K^{\text{inv}}(C_K)$ to any latent variable set $z_{\psi_0}, ..., z_{\psi_K}$ or split it across, while achieving the same optimal likelihood, since they are independent of any intervention target. The remaining information in $s_1^{\text{inv}}(C_1), ..., s_K^{\text{inv}}(C_K)$ thereby acts the same way as the noise variable $E^{t+1}$. Thus, there exist multiple maximum likelihood solutions with different splits of information to causal factors.

However, on the other hand, the solution space is yet restricted by the assignment of $s_i^{\text{var}}(C_i)$. In particular, if $s_i^{\text{var}}(C_i)$ cannot be split further into an invariable/intervention-independent part, we can rely on the same results from the previous setting, when considering $s_i^{\text{var}}(C_i)$ as new causal variables. In case there exist another split of $C_i$ which would add more information to $s_i^{\text{inv}}(C_i)$, this part could not be guaranteed to be matched to the causal factor $C_i$ due to its independence. Hence, in conclusion here, we can guarantee that $z_{\psi_i}$ will model all information of $s^{\text{var}}(C_i)$ and no other causal factor $s^{\text{var}}(C_j), i \neq j$, if there does not exist another split of $s^{\text{var}}(C_i)$. The additional information of $E^{t+1}$ as well as $s^{\text{inv}}(C_j)$ can be assigned to any causal variable. In the third step of the proof (Appendix A.5), we discuss how one can yet obtain a unique solution.

### A.4.3 CONFOUNDED INTERVENTIONS

In the previous discussion, we have used the assumption that interventions are independent of each other: $I_i^{t+1} \perp\!\!\!\perp I_j^{t+1}|C^t$. This assumption was required for showing that conditioning information of $C^{t+1}$ on any other target will lead to the same entropy as having it without a target, *i.e.* $H\left(C_i^{t+1}|C^t, I_j^{t+1}\right) = H\left(C_i^{t+1}|C^t\right)$. In this section, however, we consider a wider range of interventions. Specifically, we assume that the intervention targets $I_1^{t+1}, ..., I_K^{t+1}$ are confounded by some unobserved variable $R^{t+1}$ besides $C^t$. This allows the modeling of, for example, single-target interventions or groups of interventions, *e.g.* $I^{t+1} \in \{[0,0,0], [1,1,0], [0,1,1]\}$ for a three-variable case. Under such a setup, the entropy equation from before, *i.e.* $H\left(C_i^{t+1}|C^t, I_j^{t+1}\right) \neq H\left(C_i^{t+1}|C^t\right)$, is not valid anymore since $I_j^{t+1}$ and $I_i^{t+1}$ are not necessarily independent anymore and hence $C_i^{t+1} \not\perp\!\!\!\perp I_j^{t+1}$ can occur for some $i, j \in [\![1..K]\!], i \neq j$.

Despite that, a causal factor $C_i^{t+1}$ is still independent of any other target $I_j^{t+1}, i \neq j$, as long as it is conditioned on its true target and previous time step: $C_i^{t+1} \perp\!\!\!\perp I_j^{t+1}|C^t, I_i^{t+1}$. This is because $C^t$ and $I_i^{t+1}$ are all the parents of $C_i^{t+1}$, as shown in the causal graph of Figure 2. Further, suppose that there exist information of $C_i^{t+1}$ which is statistically independent of the intervention $I_i^{t+1}$, *i.e.* $s_i^{\text{inv}}(C_i) \neq \emptyset$. Then, this will also be independent of any other intervention target $I_j^{t+1}$, since $s_i^{\text{inv}}(C_i) \perp\!\!\!\perp I_i^{t+1}|C^t$, and all paths from $C_i$ to $I_j^{t+1}$ include $I_i^{t+1}$. Hence, our discussion of the intervention-independent

parts follow the same logic as in Appendix A.4.2, and we are left with showing that $s_i^{\mathrm{var}}(C_i)$ is modeled by $z_{\psi_i}$ in any maximum likelihood solution.

For this, we consider a pair of variables $C_i, C_j$, for which $I_i^{t+1} \not\perp\!\!\!\perp I_j^{t+1}|C^t$, and show under which circumstances we can guarantee that no information of $s_i^{\mathrm{var}}(C_i)$ will be modeled in $z_{\psi_j}$. It is sufficient to limit the discussion to pairs of variables, since one latent variable can be only assigned to a single causal variable, hence to the actual one it belongs to, $C_i$, or any other variable $C_j$ here. A crucial insight to the discussion will be that the influence of $I_j^{t+1}$ to $C_i^{t+1}$ solely relies on $I_j^{t+1}$ being correlated to both variables. Further, one requirement is that no additional conditional independence relations exist, such as $s_i^{\mathrm{var}}(C_i) \perp\!\!\!\perp I_i^{t+1}|I_j^{t+1}, C^t$, which is covered by our faithfulness assumption. Now, under this setup, we consider three cases:

1. for every time step $t$, the two variables $C_i, C_j$ have always been intervened on together, *i.e.* $I_i^{t+1} = I_j^{t+1}$ for any $t$;

2. there exist a time step $t$ at which $I_i^{t+1} = 0, I_j^{t+1} = 1$;

3. there exist a time step $t$ at which $I_i^{t+1} = 1, I_j^{t+1} = 0$.

Note that the only excluded case is when for every time step $t$, $I_i^{t+1} = 0, I_j^{t+1} = 0$. This case refers to not having observed interventions for any of the two variables, and goes back to Appendix A.4.2, where the variable part of $C_i$ is empty, *i.e.* $s_i^{\mathrm{var}}(C_i) = \emptyset$. Hence, in that case, $s_i^{\mathrm{var}}(C_i)$ would have no influence on the modeled solution.

In the first case, since the two factors have always been intervened on together, we know that $I_i^{t+1} = I_j^{t+1}$ for any time step $t$. Hence, one can assign the information of $s_i^{\mathrm{var}}(C_i^{t+1})$, $s_j^{\mathrm{var}}(C_j^{t+1})$, or the union of both to either intervention target $I_i^{t+1}$ or $I_j^{t+1}$, without losing any information. Moreover, if $s_i^{\mathrm{var}}(C_i^{t+1})$ has multiple independent dimensions, *i.e.* can be written as a product of multiple, conditionally independent variables, one can even split information of $C_i$ over the two targets. This shows that in the general case, we cannot disentangle between two variables which have always been intervened on together. Similarly, if more than 2 variables have always been intervened on together, we cannot disentangle among all those variables.

For the second case, we can deduce that there must be interventions provided for at least the observational case, *i.e.* $I_i^{t+1} = 0, I_j^{t+1} = 0$, the case where $C_j$ is intervened on but not $C_i$, *i.e.* $I_i^{t+1} = 0, I_j^{t+1} = 1$, and either the joint intervention on both $C_i, C_j$ or only interventions on $C_i$, not $C_j$. The reason why one of the two latter cases needs to exist is that if it would not be the case, $I_i^{t+1} = 0$ would be zero for any $t$. In that case, the minimal causal mechanism of $C_i$ uses $s_i^{\mathrm{var}}(C_i) = \emptyset$, hence making the modeling of $s_i^{\mathrm{var}}(C_i)$ irrelevant for the maximum likelihood solution.

Thus, from these different intervention settings, it is apparent that there cannot exist a deterministic function $f$ with which we can determine $I_i^{t+1}$ from seeing $I_j^{t+1}$. If we observe joint interventions on both variables, then for $I_j^{t+1} = 1$, both $I_i^{t+1} = 0$ and $I_i^{t+1} = 1$ can occur. Similarly, if we observe interventions on $C_i$ when $C_j$ is not intervened on, then both $I_i^{t+1} = 0$ and $I_i^{t+1} = 1$ can occur for $I_j^{t+1} = 0$. If both joint interventions and single interventions on $C_i$ have been observed, we cannot determine $I_i^{t+1}$ from $I_j^{t+1}$ at either $I_j^{t+1} = 0$ or $I_j^{t+1} = 1$. Since $s_i^{\mathrm{var}}(C_i^{t+1}) \perp\!\!\!\perp I_j^{t+1}|C^t, I_i^{t+1}$ by definition and $s_i^{\mathrm{var}}(C_i^{t+1}) \not\perp\!\!\!\perp I_i^{t+1}|C^t, I_j^{t+1}$ (the latter because $I_i^{t+1}$ is not a deterministic function of $I_j^{t+1}$, and therefore the dependence holds), we can write:

$$H\left(s_i^{\mathrm{var}}(C_i^{t+1})|C^t, I_i^{t+1}\right) = H\left(s_i^{\mathrm{var}}(C_i^{t+1})|C^t, I_i^{t+1}, I_j^{t+1}\right) < H\left(s_i^{\mathrm{var}}(C_i^{t+1})|C^t, I_j^{t+1}\right) \quad (24)$$

In conclusion, we cannot find the maximum likelihood solution if any information of $C_i^{t+1}$, which depends on $I_i^{t+1}$, is assigned to latent variables $z_{\psi_j}$. Hence, the maximum likelihood solution will strictly model $s_i^{\mathrm{var}}(C_i)$ in $z_{\psi_i}$.

Finally, in the third case, we can take a similar argument as for the second case. The only difference is that any of the additional intervention cases (joint or single on $C_j$), we have that from $I_j^{t+1} = 0$, both $I_i^{t+1} = 0$ and $I_i^{t+1} = 1$ can occur. Hence, the inequality in Equation (24) is still valid, and we cannot replace $I_i^{t+1}$ by $I_j^{t+1}$ for any subset of information of $s_i^{\mathrm{var}}(C_i)$. In summary, the maximum likelihood solution will strictly model $s_i^{\mathrm{var}}(C_i)$ in $z_{\psi_i}$ also for this case.

Therefore, we can summarize the results in the following statement. We can disentangle the intervention-dependent part of any two variables $C_i, C_j$, if there does not exist a deterministic function $f$ for which $I_i^t = f(I_j^t)$ holds for every time step $t$.

## A.5 STEP 3: DERIVING THE FINAL THEOREM

Now that we have discussed the class of disentanglement functions $\Delta^*$ with their corresponding solutions, we can take the final step by adding constraints that ensure a unique solution. In all the settings discussed in Appendix A.4.2 and Appendix A.4.3, the problem is that intervention-independent information can be represented in any of the latent sets $z_{\psi_0}, ..., z_{\psi_K}$ without affecting the optimal likelihood. However, our main goal in getting a causal representation is that we disentangle information from different causal factors, meaning that we want to guarantee that the latents of $z_{\psi_i}$ will only model information of the causal factor $C_i$, and no other causal factor $C_j, i \neq j$. Thus, we can do this by collecting all intervention-independent information in $z_{\psi_0}$. In other words, our ideal solution would be to have the latents of $z_{\psi_i}$ model $s_i^{\mathrm{var}}(C_i)$ ($i = 1, ..., K$), and $z_{\psi_0}$ to model $\{s_1^{\mathrm{inv}}(C_1), ..., s_K^{\mathrm{inv}}(C_K)\}$, where the split $s_i^{\mathrm{var}}, s_i^{\mathrm{inv}}$ was chosen to maximize the entropy of $p(s_i^{\mathrm{inv}}(C_i^{t+1})|C^t)$. To find both the right splits and collecting all intervention-independent information in $z_{\psi_0}$, we thus want to find the representation function $\hat{\delta} \in \Delta^*$ which maximizes the entropy of $z_{\psi_0}$ while maintaining the optimal likelihood. If any intervention-independent information would not be modeled in $z_{\psi_0}$, it implies that there must exist another solution with greater entropy in $z_{\psi_0}$, since all $s_1^{\mathrm{inv}}(C_1), ..., s_K^{\mathrm{inv}}(C_K)$ as well as $s_1^{\mathrm{var}}(C_1), ..., s_K^{\mathrm{var}}(C_K)$ are conditionally independent of each other (*i.e.* adding parts to $z_{\psi_0}$ cannot reduce the entropy). Further, since we try to maximize the entropy of $z_{\psi_0}$, we find the information splits $s_i^{\mathrm{var}}, s_i^{\mathrm{inv}}$ that maximize the entropy of its intervention-independent part. This is the same split as we had defined as minimal causal mechanisms in Appendix A.4.1. Thus, we can summarize this result as follows:

**Theorem A.4.** *Suppose that $\phi^*$, $\theta^*$ and $\psi^*$ are the parameters that, under the constraint of maximizing the likelihood $p_\phi(g_\theta(x^{t+1})|g_\theta(x^t), I^{t+1})$, maximize the entropy of $p_\phi(z_{\psi_0}^{t+1}|z^t, I^{t+1})$. Then, with sufficient latent dimensions, the model $\phi^*, \theta^*, \psi^*$ learns a latent structure where $z_{\psi_i}^{t+1}$ models the minimal causal variable of $C_i$ if $C_i^{t+1} \not\perp\!\!\!\perp I_i^{t+1}|C^t, I_j^{t+1}$ for any $i \neq j$. All remaining information independent of any interventions is modeled in $z_{\psi_0}$.*

The conditional independence $C_i^{t+1} \not\perp\!\!\!\perp I_i^{t+1}|C^t, I_j^{t+1}$ ensures that there exists no deterministic function $f$ for which $I_i^{t+1} = f(I_j^{t+1})$. This also includes when $I_i^{t+1}$ is constant, *i.e.*, when $C_i$ is intervened all the time or not at all, since then, $C_i$ becomes independent of $I_i^{t+1}$.

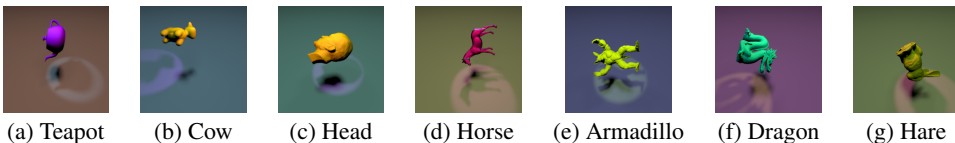

| (a) Teapot | (b) Cow | (c) Head | (d) Horse | (e) Armadillo | (f) Dragon | (g) Hare |

Figure 5: An example image for each object shape in the Temporal-Causal3DIdent dataset.

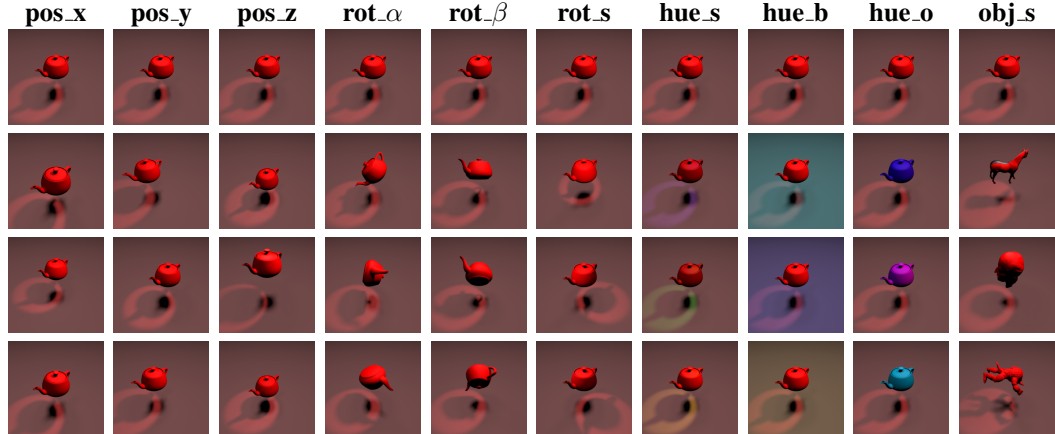

Figure 6: Visualizing the different factors of variation in the Temporal Causal3DIdent dataset. Each column represents one dimension of a causal factor, and the different rows show the original image (first row) with only the corresponding causal factor being changed.

## B    TEMPORAL CAUSAL3DIDENT DATASET

The creation of the Temporal Causal3DIdent dataset closely followed the setup of von Kügelgen et al. (2021). We used the code provided by Zimmermann et al. (2021)[1] to render the images via Blender (Blender Online Community, 2021). We will publish the adapted code for this dataset generation as well as the full datasets used here upon publication with an accompanying license.

### B.1    CAUSAL FACTOR DESCRIPTION

To begin with, we give a more detailed description of the 7 causal factors here, and provide examples of varying individual factors in Figure 6:

- The **object position** (pos_o) is modeled in 3 dimensions ($x$ - depth dimension, $y$ - horizontal, $z$ - vertical). All values are scaled between -2 as minimum and a maximum of 2, following Zimmermann et al. (2021). For the $y$ and $z$ dimension, this ensures that the object stays within the frame. For the $x$ dimensions, it ensures that the object does not cover the whole camera image, but also does not become too small in resolution for recognizing rotations and shapes.

- The **object rotation** (rot_o) is modeled in 2 dimensions ($\alpha$ - roll angle, $\beta$ - pitch angle). All dimensions use circular values of $[0, 2\pi)$, meaning that in distributions, we consider values close to 0 and $2\pi$ as close as well. The rotation is restricted to two dimensions to guarantee that every object rotation has a unique value assignment of the angles. This can be violated when modeling three angles with a value range of $[0, 2\pi)$.

- The **spotlight rotation** (rot_s) is the positioning of the spotlight as an angle. The value range is $[0, 2\pi)$, where, similarly as the object rotation, we consider it to be circular.

- The **spotlight hue** is the color of the spotlight. The value range is, again, $[0, 2\pi)$, where 0 corresponds to red. Note that the color appearance of the spotlight changes with the object and background color, since we only see the combined reflected color.

---

[1] https://github.com/brendel-group/cl-ica

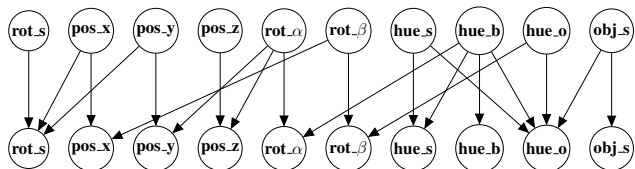

Figure 7: Causal relations between all dimensions of the causal factors of the Temporal Causal3DIdent dataset. The causal graph of Section 4 summarizes pos_x, pos_y, pos_z into pos_o and rot_$\alpha$, rot_$\beta$ into rot_o. See Appendix B.3 for details on the conditional distributions.

- The **background hue** (hue_b) is the color of the background. The value range is $[0, 2\pi)$ with the same color spectrum as the spotlight hue.

- The **object hue** (hue_o) is the color of the object, and follows the same setup as the background hue.

- The **object shape** (obj_s) is a categorical variable describing the object shape. For the 7-shape dataset version, we consider the same object shapes as von Kügelgen et al. (2021): Cow (Crane, 2021), Head (Rusinkiewicz et al., 2021), Dragon (Curless & Levoy, 1996), Hare (Turk & Levoy, 1994), Armadillo (Krishnamurthy & Levoy, 1996), Horse (Praun et al., 2000), Teapot (Newell, 1975). An example image for each of the objects is shown in Figure 5.

## B.2 DATASET GENERATION

The datasets are generated by starting at a random sample of all causal factors. For each next time step, we generate a sample according to the conditional distributions of each causal factor (see Appendix B.3 for details on the distributions). Additionally, we sample intervention targets $I_1^{t+1}, ..., I_7^{t+1}$ for all 7 causal factors. For the datasets in Section 4, we sample the targets from $I_i^{t+1} \sim$ Bernoulli(0.1). For each causal factor for which the intervention target is one, we replace its previously sampled value with a new value randomly sampled from a uniform distribution. For angles and hues, the distribution is $U(0, 2\pi)$, while for the positions, we use $U(-2, 2)$. For the object shape, we uniformly sample one out of the seven shapes. After performing the interventions, the sampled vector of causal factors is used to generate an image of a resolution of $64 \times 64$ using Blender. Note that for visualization purposes, the depicted images in this section are shown in higher resolution ($256 \times 256$). This makes it easier to recognize the object shapes and their rotations. However, we use a resolution of $64 \times 64$ in experiments to keep the computational cost of the experiments in a reasonable range.

Repeating this generation procedure for several steps results in one long sequence, which we use as a dataset. For the experiments on Temporal-Causal3DIdent Teapot, we generate a sequence of 150,000 images. For the experiments on Temporal-Causal3DIdent 7-shapes, the chosen dataset size was 250,000 images. The large dataset sizes were chosen to prevent any sampling bias and focus the experiments on general identifiability. We noticed that smaller dataset sizes such as 50,000 images still gave good scores on the correlation metrics, but a decrease in the triplet evaluation was noticeable for most causal factors, especially the position and rotation.

## B.3 TEMPORAL CAUSAL RELATIONS

Below, we define the transition functions used in the causal graph of Temporal Causal3DIdent dataset (see Figure 7 for the relations on individual causal dimensions). The chosen function forms are inspired by the ones defined by von Kügelgen et al. (2021) for the Causal3DIdent dataset.

For the position, rotation, and hue values, we sample new values over time with the following

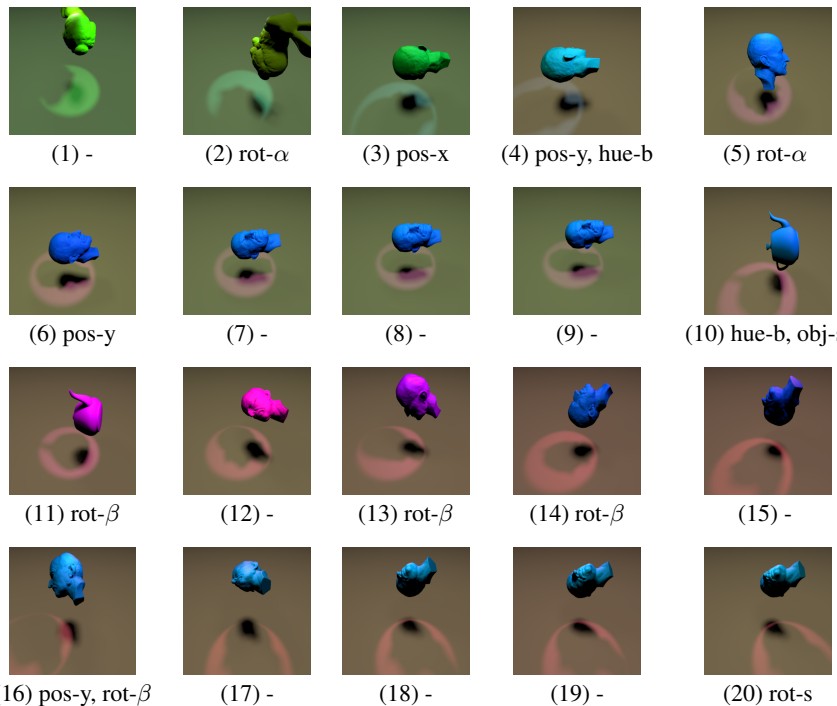

Figure 8: An example sequence with 20 frames in the Temporal Causal3DIdent 7-shapes dataset with higher resolution (from left to right, top to bottom). The causal variables denoted below each image indicate the variables which were intervened on at this time step. For instance, when transitioning from the first to the second image, all variables were sampled according to their temporal dependency except rot-$\alpha$.

functions:

$$\text{pos\_x}^{t+1} = f\left(1.5 \cdot \sin(\text{rot\_}\beta^t), \text{pos\_x}^t, \epsilon_x^t\right) \tag{25}$$

$$\text{pos\_y}^{t+1} = f\left(1.5 \cdot \sin(\text{rot\_}\alpha^t), \text{pos\_y}^t, \epsilon_y^t\right) \tag{26}$$

$$\text{pos\_z}^{t+1} = f\left(1.5 \cdot \cos(\text{rot\_}\alpha^t), \text{pos\_z}^t, \epsilon_z^t\right) \tag{27}$$

$$\text{rot\_}\alpha^{t+1} = f\left(\text{hue\_b}^t, \text{rot\_}\alpha^t, \epsilon_\alpha^t\right) \tag{28}$$

$$\text{rot\_}\beta^{t+1} = f\left(\text{hue\_o}^t, \text{rot\_}\beta^t, \epsilon_\beta^t\right) \tag{29}$$

$$\text{rot\_s}^{t+1} = f\left(\text{atan2}(\text{pos\_x}^t, \text{pos\_y}^t), \text{rot\_s}^t, \epsilon_{rs}^t\right) \tag{30}$$

$$\text{hue\_s}^{t+1} = f\left(2\pi - \text{hue\_b}^t, \text{hue\_s}^t, \epsilon_{hs}^t\right) \tag{31}$$

$$\text{hue\_b}^{t+1} = \text{hue\_b}^t + \epsilon_b^t \tag{32}$$

where $f(a,b,c) = \frac{a-b}{2} + c$, and all $\epsilon$-variables being independent samples from a Gaussian distribution with standard deviation $0.1$ for positions, and $0.15$ for angles and hues. Intuitively, the function $f$ represents that we create a 'goal' position for each variable based on its parents, and move towards the goal by taking the average between goal and current position, with additive noise. This gives us a simulation of a moving system, which, however, also permits large changes without interventions.

The function of the object hue depends on the categorical object shape, which is outlined in Table 3. We use the same setup as von Kügelgen et al. (2021), where the hare is trying to blend into the background and spotlight, while the dragon tries to stand out. The colors of the other objects are spread out across the color ring.

Finally, for the object shape, we use a noisy identity map over time. With a probability of 5%, we change the current object shape with a newly sampled one from a uniform distribution. This introduces additional noise to the object shape besides the interventions.

To showcase the dependency among causal variables, we plot the marginal distribution of tuples of

| Object shape | Object hue goal |
|---|---|
| Teapot | $0$ |
| Armadillo | $\frac{1}{5} \cdot 2\pi$ |
| Hare | $\text{avg}(\text{hue\_spot}, \text{hue\_back})$ |
| Cow | $\frac{2}{5} \cdot 2\pi$ |
| Dragon | $\pi + \text{avg}(\text{hue\_spot}, \text{hue\_back})$ |
| Head | $\frac{3}{5} \cdot 2\pi$ |
| Horse | $\frac{4}{5} \cdot 2\pi$ |

Table 3: The causal relation of the object shape, background hue, and spotlight hue to the object hue goal $g$ with which we determine the next step as $\text{hue\_o}^{t+1} = f\left(g, \text{hue\_o}^t, \epsilon_{ho}^t\right)$. The angle mean is defined as $\text{avg}(\alpha, \beta) = \text{atan2}\left(\frac{\sin(\alpha)+\sin(\beta)}{2}, \frac{\cos(\alpha)+\cos(\beta)}{2}\right)$.

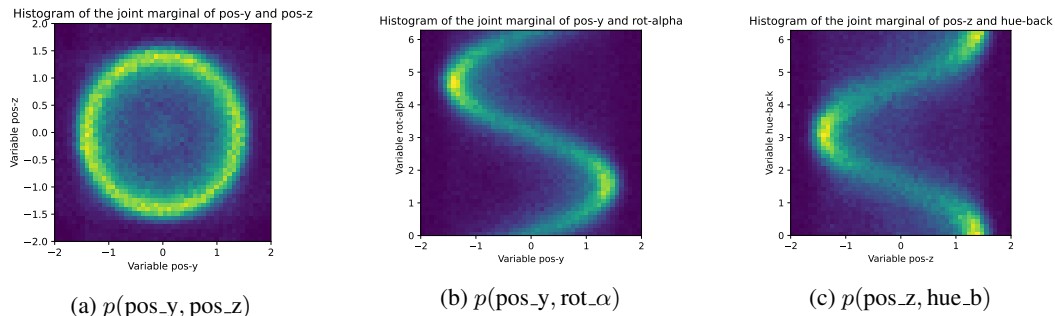

(a) $p(\text{pos\_y}, \text{pos\_z})$    (b) $p(\text{pos\_y}, \text{rot\_}\alpha)$    (c) $p(\text{pos\_z}, \text{hue\_b})$

Figure 9: Overview of the joint marginal distributions of selected variables in the Temporal Causal3DIdent dataset, showcasing the correlations among variables. The figures show histograms over the two variables, where yellow indicates a high likelihood/frequency, while dark blue has close to zero samples. (a) The two causal variables pos\_y and pos\_z share a common confounder, rot\_$\alpha$, which causes them to follow a circle with radius 1.5. (b) rot\_$\alpha$ is a parent of pos\_y, and one can see that the marginal closely follows its functional form (see Appendix B.3). (c) The hue of the background is an ancestor of pos\_z, with rot\_$\alpha$ in between the two variables. Yet, the marginal clearly follows the cosine signal, showing that the correlation goes beyond parents.

variables in Figure 9. The distributions are plotted based on a histogram of a dataset with 250,000 samples. Despite the occasional interventions, a clear correlation among variables with a confounder and ancestor-descendant relations can be seen. Overall, this shows that the chosen functions in Equation 25 to 32 introduce strong correlations among variables, which makes disentangling the factors a difficult task.

