# OpenReview forum: "CITRIS: Causal Identifiability from Temporal Intervened Sequences"
_ICLR.cc/2022/Workshop/OSC — ICLR2022 OSC  Poster_

### Official Review · Reviewer_zNfc · 2022-03-10
**A variational framework for discovering latent causal factors**

**Rating:** 3
**Confidence:** 2

**Review:**

In this work the authors propose an interesting variational framework, CITRIS, that allows to learn causal representations from temporal, interventional data, where a key focus is on representing causal factors as multi-dimensional vectors. The work focuses on the setting of temporal intervened sequences and first provides a formalisation on how to identify the “minimal causal variables” in a dataset. Finally, CITRIS thus learns an assignment of latent variables to causal factors.
The authors further provide valuable experimental results, among other things via the help of normalising flows, indicating that VAEs trained within the CITRIS framework produce a more disentangled latent space. Lastly, initial experiments also indicate good generalisation abilities to unseen data objectives. A note here is however, that the results of CITRIS are not compared to baseline VAE frameworks for the generalisation experiment making it more difficult to interpret these results.

Overall the paper is well written and the ideas well presented. The mathematical foundation, to my knowledge, appears sound. The initial experimental results look promising and the overall problem tackled in this work is very fitting for the workshop.

---

### Official Review · Reviewer_s9pT · 2022-03-16

**Rating:** 2
**Confidence:** 2

**Review:**

This paper proposes a method to learn causal temporal representations which support multidimensional causal factors (instead of single scalar factors as previously the case in the literature). This requires knowing which causal variable the interventions act on, but not their exact values/effect.

Overall, I found this paper quite interesting, although it could be more focused as it tries to present a few too many things at once (it presents some theory about minimal causal variable identification, a variational implementation using a temporal VAE architecture, as well as another variation which uses a Normalizing Flow. I might have tried to only select 2 our of the 3).

Given the focus of the workshop, this work is clearly in-scope and the ability to identify multidimensional factors is quite promising, so I’d recommend acceptance.

Comments and questions:
1. What would happen if the interventions weren’t known and used? The Introduction presents this fact as a characteristic difference to related work, but isn’t it a weakness?
   1. What would need to change to try to do the structure learning of the causal variables I as well?
2. For the Normalizing Flow version, did you explore ways to keep fine-tuning the pretrained Autoencoder?
   1. It is interesting that you can make it work using a frozen AE, but I can imagine this has the usual harsh requirements on the capabilities of the frozen model (e.g. it has to be a “good enough” model and if it does not capture specific causes by itself there is no way to salvage it).
   2. This section of the work feels a bit less explored than the rest, so it might be pushed to the Appendix for the time being?

---

### Decision · Program_Chairs · 2022-03-24

Accept (Poster)